# SUMOylation and Major Depressive Disorder

**DOI:** 10.3390/ijms23148023

**Published:** 2022-07-21

**Authors:** Seok-Won Jeoung, Hyun-Sun Park, Zae Young Ryoo, Dong-Hyung Cho, Hyun-Shik Lee, Hong-Yeoul Ryu

**Affiliations:** 1BK21 FOUR KNU Creative BioResearch Group, School of Life Sciences, College of National Sciences, Kyungpook National University, Daegu 41566, Korea; swjeoung96@gmail.com (S.-W.J.); jaewoong64@knu.ac.kr (Z.Y.R.); dhcho@knu.ac.kr (D.-H.C.); leeh@knu.ac.kr (H.-S.L.); 2Brain Science and Engineering Institute, Kyungpook National University, Daegu 41566, Korea; 3Department of Biochemistry, Inje University College of Medicine, Busan 50834, Korea; hspark@inje.ac.kr

**Keywords:** SUMOylation, major depressive disorder (MDD), neuron, synapse, mitochondria

## Abstract

Since the discovery of the small ubiquitin-like modifier (SUMO) protein in 1995, SUMOylation has been considered a crucial post-translational modification in diverse cellular functions. In neurons, SUMOylation has various roles ranging from managing synaptic transmitter release to maintaining mitochondrial integrity and determining neuronal health. It has been discovered that neuronal dysfunction is a key factor in the development of major depressive disorder (MDD). PubMed and Google Scholar databases were searched with keywords such as ‘SUMO’, ‘neuronal plasticity’, and ‘depression’ to obtain relevant scientific literature. Here, we provide an overview of recent studies demonstrating the role of SUMOylation in maintaining neuronal function in participants suffering from MDD.

## 1. Introduction

Synaptic plasticity is a process involved in the change of synaptic strength through specific patterns of synaptic activity. Changes in synaptic plasticity (also known as Hebbian plasticity) are known to modify thoughts, feelings, behavior, memory, and brain volume [1]. Long-term potentiation (LTP) and depression (LTD) are the best-studied forms of synaptic plasticity. LTP is defined as a long-term increase in synaptic strength and functionality caused by certain patterns of synaptic activity, such as increased postsynaptic α-amino-3-hydroxy-5-methyl-4-isoxazolepropionic acid receptors (AMPARs), whereas LTD is defined as a long-term decrease in synaptic strength and function [2]. As reviewed by Richer-Levin and Xu, stress has been shown to prompt LTP and TLD alterations in neurons in the CA1 region of the hippocampus [3]. Other studies have reported negative alterations in rodent CA3 neuron dendrites after exposure to chronic stress, further strengthening the theoretical link between stress, synaptic plasticity, and major depressive disorder (MDD) [4]. Synaptic plasticity is largely controlled by the activation level, location, and state of AMPARs and N-methyl D-aspartate (NMDARs). AMPARs and NMDARs are non-selective ionotropic glutamate receptors responsible for relaying excitatory synaptic transmission through the nervous system [5]. During early LTP, the synapse needs to adapt to rapid changes and thus relies on the movement and control of pre-existing proteins, activation of NMDARs, and trafficking of AMPARs [6,7,8,9]. During late-LTP, the control of gene expression and RNA translation is much more significant [10,11]. There are many reports of synaptic functions being regulated by postsynaptic glutamate receptor modifications such as phosphorylation and ubiquitination. CaMKII is known to cause the phosphorylation of Ser831 of the GluA1 subunit of AMPAR, leading to the increase in channel conductivity, which causes expression of LTP in the hippocampus [12]. The ubiquitin-proteasome system, which is responsible for the targeting and degradation of proteins, has been shown to relocate into dendritic spines during synaptic activation to alter the levels of synaptic proteins [13]. Larger-scale alterations caused by post-translational modifications (PTMs) have also been reported. Applying stress in rodent models via various methods, such as predator and social defeat stress, resulted in a sharp increase in H3 phosphorylation and acetylation in the hippocampus. Predator stress is induced by introducing natural predator odors to rodents, and social stress is initiated by placing a male rodent in the cage of a more dominant male [14,15]. It has been shown that H3 PTMs in the nucleus accumbens can alter susceptibility and resilience to stress. Mice with increased levels of H3 acetylation were shown to be more resilient to stress and subsequent depression-like phenotype, whereas increased levels of H3K9me2 caused the mice to be more susceptible to stress [16,17]. The observation of LTP-related genes shows rapid methylation and demethylation following LTP induction [18]. PTMs play significant roles in the maintenance of stress and depression.

MDD, as defined by the American Psychiatric Association, is a common and serious mental disorder accompanied by the loss of interest and pleasure in enjoyable activities (anhedonia), changes in appetite, loss of energy, insomnia, psychomotor retardation, and suicidal thoughts [19]. MDD can be potentially fatal, with an estimated one in six people having experienced depression once during their lifetime [20,21,22]. The exact causes of MDD remain elusive, with possibilities ranging from neural biochemistry to personality [23,24]. The lack of biomarkers, inconclusive genetic studies, and personal effects such as stress all contribute to the poor understanding of MDD, which also plays a role in antidepressant studies that have led to disheartening results [25,26,27]. Serotonin selective reuptake inhibitors (SSRIs), the most widely prescribed antidepressant drugs, are ineffective in up to 45% of participants even after weeks to months of drug treatment [22,28,29,30]. Although the exact mechanisms of MDD pathology remain unclear, global efforts have been made to clarify the cellular mechanisms leading to MDD. Post-mortem studies of depressed participants have shown a reduced number of prefrontal cortex (PFC) synapses, which have been logically connected to reduced synaptic functions [31]. Furthermore, the volumes of the PFC and hippocampus have been observed to shrink in individuals with depression, with the extent of shrinkage positively correlated to the severity of depression the person is experiencing [32,33].

Small ubiquitin-like modifier (SUMO)ylation is a relatively new form of PTM that is currently under careful investigation by scientists worldwide. The first mention of the direct control of the synapse by SUMOylation was in 2009, by Feligioni et al. By increasing and decreasing presynaptic SUMOylation levels in synaptosomes, they discovered fluctuations in Ca^2+^ influx. Increasing levels of SUMOylation resulted in reduced Ca^2+^ influx and decreased glutamate release, whereas the opposite was observed by reducing presynaptic SUMOylation levels [34]. In 2014, SUMOylation was demonstrated to be required for LTP activation. Using inactive forms of SUMO-conjugating enzymes (Ubc9), LTP was shown to be significantly reduced, and increasing sentrin/SUMO-specific protease 1 (SNEP1) levels showed the same effect in CA1 hippocampal slices [35]. Compared to other forms of PTMs, such as acetylation and methylation, the mechanisms and effects of SUMOylation are still elusive, but they seem to have a noteworthy effect on synaptic plasticity and overall neuronal health. Here, we review the regulation of neuronal function by SUMOylation of proteins associated with various methods of synaptic plasticity control connected to the pathogenesis of MDD.

## 2. SUMOylation

SUMO proteins are a family of proteins that are conjugated to lysine residues on target proteins. SUMO is ligated to a ΨKX(D/E) consensus motif of the target proteins, where Ψ represents a hydrophobic residue, K is a target lysine, X is any residue, and D/E represents acidic residues [36]. There are five isoforms of SUMO found in humans, SUMO1–5. While SUMO4 and 5 still require further investigation, the functions of SUMO 1~3 have been extensively investigated. SUMO1 is involved in monoSUMOylation (attachment of SUMO to a single lysine residue) and regulates protein localization and ubiquitination. SUMO2 and 3 are nearly indistinguishable and are thus referred to as SUMO2/3. SUMO2/3 is involved in polySUMOylation, forming SUMO chains on target proteins, and seems to respond to cellular stresses such as heat shock, DNA damage, and nutrient deficiency [37,38]. SUMOylation occurs in a cycle (Figure 1) and relies on the activation of four types of enzymes known as the Ulp/SENP family, E1 SUMO-activating, E2 SUMO-conjugating, and E3 SUMO-ligase. The SUMO peptide is first matured for conjugation by cleaving the C-terminus to reveal a glycine-glycine motif. The E1 enzyme, an SAE1/2 heterodimer, binds to the exposed motif, transferring SUMO to the E2 enzyme Ubc9. The E3 enzyme then transfers SUMO from Ubc9 to the target protein, completing SUMOylation [39]. There are three main E3 ligase families: SP-RING ligases, which include the Siz and PIAS family; RAN binding protein 2 (RanBP2), which is a part of the nuclear pore complex; and human polycomb protein 2 (hPc2) [40]. SUMOylation is highly dynamic; thus, SUMO peptides can be removed from the substrate by Ulp/SENP proteases, allowing SUMO peptides to be recycled [39]. There are six known members of the SENP protease family, SENP1–7 (excluding SENP4), each with specific functions and localizations. SENP1 functions in removing SUMO1 preferentially over SUMO2/3, whereas other members specialize in removing SUMO2/3 [41,42]. Three other SUMO proteases have been recently discovered, but their functions have not been extensively investigated [43]. SUMOylation can affect target proteins by blocking protein-protein interaction sites, providing binding sites for other regulatory factors, controlling protein stability, or causing conformational changes in target proteins. These changes by SUMOylation have numerous effects, including regulating transcription, cell cycle progression, DNA repair, protein trafficking, and mitochondrial dynamics, among many other functions [44,45,46,47,48,49,50,51,52]. Because of these effects, SUMOylation has been of interest to neurobiologists worldwide. SUMO1 and SUMO2/3 have been detected in all parts of the neuron, from the nucleus to the synapse. In conjunction, SUMO isoforms, Ubc9 SENPs, and PIAS proteins have been detected in synaptosomes, implying that SUMOylation may have a role in regulating neurotransmitter release and general synaptic function [53,54].

## 3. SUMOylation in the Synapse

Synaptic plasticity is associated with many cellular mechanisms, such as protein trafficking, translation, and transcription [55]. LTP and LTD are mediated through NMDAR and subsequent AMPAR translocation and activity, mainly through the Ca^2+^ influx caused by NMDARs. Increased levels of cellular Ca^2+^ lead to the activation of multiple signal transduction pathways, including the neuronal nitric oxide synthase/nitric oxide (nNOS/NO) signaling and Ras-ERK pathways. In addition to NMDARs and AMPARs, G-protein coupled receptors and serotonin receptors have also been shown to be instrumental in plasticity regulation. In this section, we summarize current studies regarding the effects of SUMOylation on specific proteins and pathways.

### 3.1. AMPARs

AMPARs are glutamate-gated ion channels responsible for rapid excitatory synaptic transmission in the central nervous system. AMPARs are comprised of four subunits that form ion channels with different functions [56]. To induce NMDAR-mediated LTP, AMPARs must be concentrated in the postsynaptic membrane. It is widely believed that this trafficking is possible due to a large number of PTM-induced modifications on AMPAR subunits, including glycosylation, palmitoylation, and phosphorylation [57]. However, a bacterial SUMOylation assay of neuronal proteins to test for SUMOylation core motifs has shown that AMPAR subunits are not SUMOylated [58]. When inducing LTP in cultured hippocampal slices, an increase in SUMO2/3 was observed, stipulating a connection between LTP induction and SUMOylation [35]. When LTP was induced in cultured hippocampal slices, an increase in SUMO2/3 was observed, indicating a connection between LTP induction and SUMOylation [25]. Furthermore, increasing surface level AMPARs by chemically inducing LTP caused an increase in SUMO1, Ubc9, and SUMO1 RNAs; however, overexpression of SENP1 prevented these increases [59]. Additionally, by silencing SUMO1-3 expression in mice, malfunctions in episodic and fear memory conditioning have been observed [60]. Synaptic plasticity plays an imperative role in fear memory conditioning; thus, SUMOylation is required for the proper trafficking of AMPARs. An AMPAR trafficking-related factor that has been identified as SUMOylated is an activity-regulated cytoskeleton-associated protein (Arc) (Figure 2). Arc levels increase rapidly in response to neuronal activity and have been shown to increase during periods of stress in rodent models [61]. Arc has been shown to participate in AMPAR trafficking and ensuing LTP and LTD induction [62,63]. Arc has been reported to be SUMOylated at lysine residues 110 and 268, and SUMOylation of these residues regulates protein-protein interactions and synaptic scaling [64]. In vivo studies have shown that Arc SUMOylation is involved in LTP induction and vice versa; LTP induction gives rise to Arc SUMOylation, which is repressed when LTP is inhibited [65].

### 3.2. nNOS/NO Signaling

Nitric oxide is a secondary messenger found in mammalian cells and is produced by NOS enzymes. nNOS is an NOS isoform found in neurons and is important for proper neuronal function. Post-mortem studies of MDD participants have found increased levels of nNOS expression in the hippocampus, whereas inhibition of nNOS prevented depression-like behavior in rodent models exposed to chronic mild stress [66,67]. A study by Du et al. has shed light on nNOS SUMOylation, showing that nNOS is SUMOylated at lysine residues 725 and 739. SUMOylation of these residues promotes nNOS phosphorylation, which increases NO production (Figure 2). By disrupting nNOS pathways, activity-induced nNOS SUMOylation also decreases LTP-related Arc expression and blocks ERK 1/2 signaling [68].

### 3.3. Ras/Raf/MEK/ERK Pathway

The Ras/Raf/mitogen-activated protein kinase/ERK kinase (MEK)/extracellular signal-regulated kinase (ERK) signaling cascade (shortened to the ERK pathway) relays signals from cell surface receptors to transcription factors to regulate gene expression [69]. The ERK pathway has been the most studied signal transduction pathway, with many reports of decreased ERK signaling in the PFC and hippocampus of suicidal MDD participants [70,71,72]. Decreased MEK-ERK signaling was detected in the hippocampus of suicidal participants, while MAPK phosphatase (MKP), a MAPK inhibitor, was increased [73]. Duric et al. (2010) discovered increased levels of MKP in mice undergoing chronic unpredictable stress, with ERK1/2 signaling decreased in the PFC and hippocampus, while deletion of the *Mkp-1* gene resulted in resistance to stress [72]. Ras activation has also been implicated in the trafficking of AMPARs during LTP [74]. All isoforms of Ras are SUMOylated at lysine 42, and SUMOylation of K42 is crucial for Ras-promoted signaling, as the K42R mutation causes reductions in downstream activation of the ERK signaling cascade (Figure 2) [75,76].

Downstream of the ERK signaling cascade, cAMP-responsive element-binding protein (CREB) has also been shown to decrease in the hippocampus of MDD participants [72]. As a phosphorylation target of the ERK signaling cascade, CREB is a transcription factor required for the transcription of genes that regulate neuronal plasticity. Chronic stress models have also been used to test these results; indeed, chronic stress leads to lower CREB activity in rodent models, indicating that stress can lower CREB levels in the brain, which could lead to MDD development [77,78]. By increasing levels of PIAS1 or NMDA in rodent models, a research team has shown enhanced SUMOylation of CREB in the hippocampus. They demonstrated two predominant lysine residues that are SUMOylated (lysine 271 and 290). Interestingly, preventing SUMOylation increases the phosphorylation of CREB, while preventing CREB phosphorylation inhibits CREB SUMOylation. These data imply cross-talk between the two types of PTMs and require further research [79]. In addition to the above-mentioned effects, CREB SUMOylation also increases the transcription levels of brain-derived neurotrophic factor (BDNF), which we will discuss in greater detail in the next section of this review.

### 3.4. BDNF

BDNF is a growth factor involved in many central functions of the brain, including synaptic plasticity [80]. BDNF functions by binding to tropomyosin receptor kinase B (TrkB) in the cellular membrane and triggering a signaling cascade that activates various pathways, including the ERK cascade and CREB [81]. Thus, BDNF has been studied with great interest regarding mood disorders, and some antidepressants act upon BDNF-TrkB [82]. Environmental stress that triggers depression can lower BDNF mRNA levels [83]. While BDNF has been thoroughly studied in other fields, SUMOylation of BDNF has not yet been elucidated. A recent study showed that BDNF-TrkB signaling can alter the subcellular localization of SUMOylation enzymes. PIAS3 is functionally regulated and translocated from the nucleus to the dendrites via the ERK1/2 pathway, which is downstream of BDNF-TrkB [84].

A downstream event of BDNF-TrkB signaling is the activation of the P13K-Akt pathway. In MDD participants, reduced levels of P13K-Akt were observed in the PFC, while antidepressant treatment was shown to increase Akt signaling [85,86]. Akt1 is SUMOylated at lysine 64 and 276 by PIAS3 following NMDAR activity. SUMOylation of these residues increases Akt enzymatic activity, and inactivation of PIAS3 impairs LTP expression [87].

### 3.5. SNARE Complex

In yeast, mammalian, and plant cells, vesicle fusion with the cellular membrane is mediated by a complex of proteins in the soluble N-ethylmaleimide-sensitive factor attachment protein receptor (SNARE) protein family, commonly known as the SNARE complex [88]. SNAREs have been thoroughly studied in the brain, as they mediate neurotransmitter release through synaptic vesicles. SNARE proteins are categorized into synaptosomal-associated proteins (SNAP), syntaxins (stx), and target SNAREs (t-SNAREs), depending on their location within the synapse [89]. Synapsin 1a (Syn1a) is a protein that regulates vesicle availability at the pre-synapse. There is evidence showing that Syn1a is SUMOylated at lysine 687 and mutations to K687R result in a decrease in available releasable vesicles and impaired exocytosis, indicating SUMOylation as a Syn1a function regulator [90]. Stx1a is another member of the SNARE family, which is involved in the release of neurotransmitters and the recycling of synaptic vesicle membranes and proteins from the plasma membrane [91]. Upon NMDAR activation, Stx1a is SUMOylated at three lysine residues (K252, K253, and K256), and mutations of the three lysine residues to arginine cause an imbalance in synaptic vesicle endocytosis and exocytosis, indicating that Stx1a SUMOylation is a regulator of neurotransmitter release [92]. Rab3-Interacting molecule 1α (RIM1α) regulates vesicle release from synapses. Further research has shown that RIM1α plays a crucial role in the docking of synaptic vesicles [93]. Studies have shown that RIM1α is SUMOylated at lysine 502, which aids in the clustering of Cav2.1 calcium channels. K502R mutations in RIM1α decrease synaptic exocytosis [94].

### 3.6. Ion Channels

Neuronal excitability is maintained by the properties and functionality of ion channels found in the plasma membranes of neurons. Voltage-dependent potassium channels (Kv) are gated neuronal transmembrane channels and are the most abundant type of ion channel. Kvs are essential for action potential generation and are phosphorylated during LTP to increase neuronal excitability [95]. There have been several reports of Kv SUMOylation. Kv1.1 was modified by SUMO1/2 and interacted with SENP2 in hippocampal neurons. Through SENP2 knock-out, hyperSUMOylation was observed on Kv1.1, Kv7.2, and Kv7.3 in rodent model hippocampal neurons. Kv1.1 hyperSUMOylation did not appear to have any effect on channel activity. Kv7.2 and 7.3 SUMOylation resulted in a diminished M-current, which is responsible for neuronal hyperexcitability. Neuronal hyperexcitability is known to cause epilepsy, neuropathic pain, and degeneration [96,97]. Kv1.5 SUMOylation regulates channel inactivation, while K4.2 is SUMOylated at two lysine residues, K437 and K579, which increase Kv4.2 surface expression while decreasing current conductivity [98,99]. Similar effects have been observed by SUMOylation of the Kv2.1, Kv7.1, and Kv11.1 channels [100,101,102]. However, studies on sodium channels have been scarce, with only one study implicating the results of direct voltage-gated sodium channel (NaV) SUMOylation. SUMOylation of NaV1.2 resulted in increased current conductivity. Collapsin response mediator protein 2 (CRMP2) has been previously shown to regulate NaV1.7 channels and has been reported as a SUMOylation target. A K374A mutation in CRMP2 causes a reduction in NaV1.7 current conductivity by interfering with CRMP2 and NaV1.7 interactions [103,104].

### 3.7. Receptors

#### 3.7.1. Serotonin Receptors

Changes in serotonin receptors are a critical step in MDD development. There are seven classes of serotonin receptors, each with a different function. Of these classes, 5-HT1A primarily regulates reward processing, mobility, appetite, and anxiety [105]. Recently, 5-HT1A was reported to be SUMOylated by SUMO1 in rodent brains. Overexpression of PIAS in cells has been shown to increase SUMOylation, while SENP2 decreases SUMOylation of 5-HT1A, although the implications of 5-HT1A SUMOylation are yet to be discovered [106].

#### 3.7.2. Dopamine Transporters

Dopamine is a neuromodulator involved in stimulus-reward learning processes and behavioral control. Dopamine functions by altering the properties of neurons, such as membrane excitability, neurotransmitter release, and protein trafficking [107]. Dopamine is regulated by dopamine transporters (DAT), which reuptake dopamine back into the pre-synapse. Overexpression of Ubc9 and SUMO1 enhanced DAT surface expression and stability, and increased DAT functionality, whereas knockdown of Ubc9 resulted in lower SUMO1-DAT levels and increased DAT degradation [108].

#### 3.7.3. G-Protein Coupled Receptors (GPCRs)

Only a handful of GPCRs are SUMOylated in eukaryotes. mGluR7 and mGluR8 are located in the presynaptic region and have different functions. mGluR7 plays a vital role in controlling excitatory synapse function, whereas mGluR8 controls Ca^2+^-dependent neurotransmitter release [106,109]. mGluR7 is SUMOylated at lysine 889, whereas mGluR8 is SUMOylated at lysine 882 and 903. SUMOylation was found to regulate the endocytosis of these receptors [110,111]. Interestingly, mGluR7 shows different SUMOylation patterns depending on the cell type. In HEK293T cells, mGluR7 is SUMOylated by both SUMO1 and SUMO2/3, whereas only SUMO1 is found in hippocampal cells [112]. Additionally, mGluR5 activation reduces the diffusion of Ubc9 out of the dendritic membrane, increasing Ubc9 levels inside neurons [113]. Similarly, SENP1 levels also increase in dendrites upon mGluR5 activation [114]. The M1 muscarinic acetylcholine receptors (M1Rs) are another type of GPCR that is SUMOylated. They mediate signals via acetylcholine and play vital roles in learning and memory. M1R is SUMOylated at lysine 237 and receptor activation removes SUMOylation at this residue. SUMOylation at this site increases the binding capacity, sequentially raising signal transduction levels. K237R mutations decrease SUMOylation and signal transduction through M1R [115]. Another GPCR that has been researched is cannabinoid receptor 1 (CB1). They function as modulators of the voltage-gated calcium and potassium channels. Activation of CB1 has been shown to increase SUMO1 conjugation and free SUMO1 levels in cortical neurons of rodent models, although the exact functions of SUMOylation are yet to be discovered [116].

#### 3.7.4. Kainite Receptors (KARs)

KARs are relatively under-studied members of the glutamate receptor family. AMPARs and NMDARs play important roles in neuronal function and the regulation of plasticity, such as regulating neurotransmitter release and membrane excitability [117]. A subunit of KAR, GluK2, is SUMOylated singly at the lysine 886 residue in rodent hippocampal neurons. KAR activation by glutamate binding caused K886 to be SUMOylated, and the KAR was internalized, which was confirmed by infusing SUMO1 into the post-synapse and detecting lower levels of KAR signals [53].

### 3.8. Cytoplasmic Polyadenylation Element-Binding Protein 3 (CPEB3)

CPEB3 exists in a soluble inactive form or as an insoluble active aggregate and regulates the synthesis of synaptic proteins. Appropriate localization and activation are crucial for the function of CPEB3 in LTP [118]. SUMOylation is necessary for the accurate localization of CPEB3 to processing bodies, implicating functions related to translation. Under basal conditions, CPEB3 is SUMOylated by SUMO2/3, which is then rapidly deSUMOylated during LTP transduction, and subsequently translocated into polysomes to aid translation [118,119]. Other reports have shown that SUMO2 conjugation may play a role in silencing the local translation after a pulse of neuronal activity. SUMOylation by SUMO2 shifts the binding affinity of mRNA from deSUMOylated CPEB3 aggregates to inactive SUMOylated CPEB3 monomers [107].

## 4. Mitochondrial Dysfunction

Commonly known as ‘the powerhouse of the cell’, mitochondria have major functions in neuronal processes, such as Ca^2+^ regulation, plasma membrane potential maintenance, the release of neurotransmitters, and maintenance of high energy levels [120]. It is not surprising that MDD is linked to mitochondrial function. Chronic mild stress induced in mice resulted in the inhibition of oxidative phosphorylation and dissipated mitochondrial membrane potential in various brain regions, including the hippocampus [121]. Glucose utilization by the PFC and other brain regions is reduced in depressed human subjects, implying that mitochondrial health is a requirement for regulating mood [122]. Mitochondrial health is balanced by the tight regulation of mitochondrial fission, fusion, and mitophagy. Fission is responsible for mitochondrial renewal, redistribution, and partitioning of damaged parts of the mitochondria, while fusion is the process of neighboring mitochondria mixing their contents to restore function after being damaged [120,123]. Mitophagy is a ‘cleaning’ mechanism that removes dysfunctional mitochondria from the cell through autophagy [124]. Although no studies have directly linked SUMO-mediated mitochondrial dysfunction to MDD, depression is a common symptom in participants with Parkinson’s disease [125].

### 4.1. Dynamin-Related Protein 1 (Drp1)

A SUMOylation target that has been extensively studied in relation to neuronal dysfunction in Parkinson’s disease is the cytosolic GTPase Drp1. Drp1 is recruited to the outer mitochondrial membrane to regulate mitochondrial fission [126,127]. Drp1 is regulated by several PTMs, including SUMOylation. Drp1 has been found to be SUMOylated at various lysine residues throughout the protein. Drp1 interacts with Ubc9 and is SUMOylated by all SUMO isoforms [128]. SUMO-1 overexpression prevents Drp1 degradation, leading to enhanced mitochondrial fission and subsequent apoptosis (Figure 3) [129]. Intriguingly, a study in 2013 by Guo et al. showed that SENP3 degradation caused increased conjugation of SUMO2/3 to Drp1, which inhibited mitochondrial fission [130]. Drp1 activity is regulated by SENPs. SENP5 overexpression causes a reduction in SUMO-1-induced mitochondrial fission, while the opposite is true with the knockdown of SENP2 and SENP5 [131,132].

### 4.2. DJ-1

There have been previous reports of increased reactive oxygen species (ROS) production in participants with depression [133]. Under oxidative stress conditions, the protein deglycase DJ-1 is translocated into the mitochondria, where it functions as a mitochondrial complex I regulator and contributes to protecting the cell from oxidative stress [134,135]. DJ-1 must be suitably SUMOylated at K130 to function. Mutations that interfere with DJ-1 SUMOylation were found to perturb the antioxidant functions of DJ-1. It was found that improper SUMOylation decreases DJ-1 solubility, leaving the protein susceptible to proteasomal degradation [136]. SENP3 regulates mitophagy by interacting with Fis1, a protein required for mitophagy. Fis1 was discovered to be SUMOylated at lysine 149, and it was found that K149 SUMOylation prevents Fis1 from causing mitophagy (Figure 3) [137]. Interestingly, DJ-1 was found to promote proteasomal degradation of Fis1 in response to oxidative stress [138]. Additionally, Fis1 recruits Drp1 from the cytosol to the mitochondrial outer membrane [139]. These results suggest an interesting response to oxidative stress, involving lower levels of mitophagy and increased fission mediated by SUMOylation.

### 4.3. Peroxisomal Proliferator-Activated Receptor-γ Coactivator 1 α (PGC-1α)

PGC-1α is a transcriptional coactivator of energy metabolism genes and a key regulator of mitochondrial biogenesis and function [140]. SUMO-1 conjugation at K138 inhibits PGC-1α transcriptional activity [141]. SENP1-mediated removal of SUMO-1 can recover the negative effects caused by SUMOylation and PGC-1α transcriptional activity [142]. PGC-1α is under the indirect control of DJ-1, mediated by pyrimidine tract-binding protein-associated splicing factor (PSF, Figure 3). DJ-1 inhibits SUMOylation of PSF, which in turn binds to PGC-1α and suppresses its transcriptional activity [143]. This demonstrates a link between oxidative stress and SUMO-mediated abnormal mitochondrial gene expression.

### 4.4. Parkin

Parkin is a ubiquitin E3 ligase, which is a crucial regulator of mitochondrial fission and fusion. Parkin can bind and ubiquitinate Drp1 to promote Drp1 degradation and prevent mitochondrial fission. This effect can be nullified by SUMOylation of parkin by SUMO-1. SUMO-1 conjugation results in parkin being translocated into the nucleus, reducing the amount available for activity in the mitochondria [144].

### 4.5. Parkin-Interacting Substrate (PARIS)

PARIS is a transcriptional repressor expression of PGC-1α. PARIS levels are regulated by the ubiquitin E3 ligase activity of parkin (Figure 3) [145]. PARIS is SUMOylated with SUMO1 and SUMO2/3 at K189 and K286, respectively, by PIASy, while the effects of SUMOylation are unclear and dependent on cell type. SUMOylation of HepG2 cells did not affect PARIS activity, while SH-SY5Y cells with 2KR PARIS mutations showed restoration of repression effects. Conversely, in HEK293 cells, 2KR mutants show increased transcriptional repression [146]. In subsequent studies, SUMOylation of PARIS with SUMO2/3 induced ubiquitination and proteasomal degradation, thereby relieving PARIS-mediated transcriptional repression [147]. Although there have been few direct connections between MDD and mitochondria-related SUMOylation, the above data show that SUMOylation plays a significant role in mitochondria-mediated neuronal health, and future studies into MDD-related mitochondrial dysfunction could reveal prospective methods of treatment.

## 5. Conclusions

Since its discovery in 1995, SUMOylation has been identified as a key regulator of many functions in eukaryotic cells. Although yet unclear, the above results do show that SUMOylation plays an integral role in the proper regulation of neural function. Recently, SUMOylation has been researched using non-neuronal models, showing promising results. An example would be the control of Fis1 localization in HeLa cells by SENP3-mediated SUMOylation or the localization of mitochondria in HEK-293 embryonic kidney cells by mitofusin1/2 mediated by SUMO2 modifications [137,148]. Mitophagy in neurons has recently been reviewed by Doxaki and Palikaras, indicating the importance of mitophagy in neuronal survival and cellular homeostasis [149]. As there are currently no studies on the effects of mitophagy-related SUMOylation on neuronal function, this seems to be a promising field for further study. In addition, chronic pain has been linked to depression in many clinical studies, with some reports stating that up to 85% of participants experiencing chronic pain are also affected by severe depression [150]. Recent studies have reported on the involvement of SUMOylation in pain regulation. One study showed that preventing SUMOylation of CRMP2, a NaV1.7 channel interacting protein, was antinociceptive in chronic and acute pain models using rodents [151]. A different study on neuropathic pain management reported that enhancing SUMOylation of DGCR8 elevated sensitivity to pain in rat models of spinal nerve ligation [152]. These results imply that targeting SUMOylation is a promising new approach for pain relief and antidepressant medication. Experiments with existing antidepressants have shown the integral role of SUMOylation. Tricyclic antidepressants inhibited FKBP51 (a glucocorticoid receptor inhibitor) SUMOylation by blocking interactions with PIAS4 and subsequently restoring glucocorticoid receptor activity [153]. Although inconclusive, recent advances have shown promising results, suggesting that targeting of SUMOylation-related enzymes or SUMOylated proteins may serve as a potential therapeutic target in the future.

## Figures and Tables

**Figure 1 ijms-23-08023-f001:**
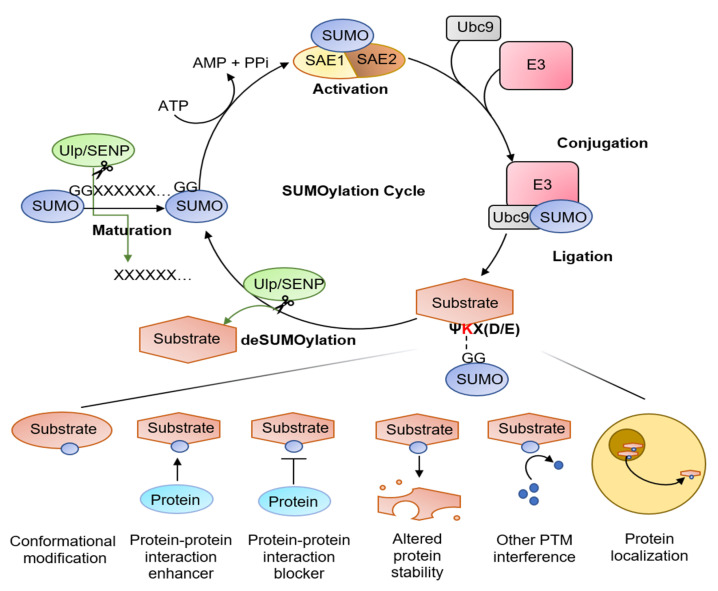
A diagram of the SUMOylation pathway. Small Ubiquitin-like MOdifier (SUMO) is first synthesized as an inactive precursor, which is subsequently cleaved by sentrin/SUMO-specific protease (SENPs) to reveal the double glycine motif and create mature SUMO proteins. SUMO is then activated by E1 proteins (SAE1 and SAE2 heterodimer). The activated SUMO protein is transferred to the Ubc9 E2 enzyme, which works in unison with E3 enzymes to target and ligate the SUMO protein onto substrates. The SUMO protein can be recycled by SENP-mediated cleavage of SUMO. SUMOylation either enhances or blocks protein-protein interactions, modifies substrate conformation, blocks sites for other post-translational modifications (i.e., ubiquitin), and aids in protein localization.

**Figure 2 ijms-23-08023-f002:**
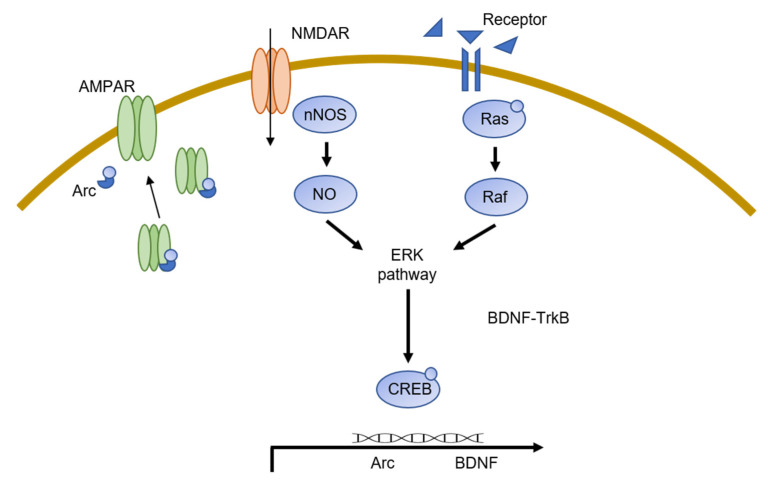
SUMOylation in the neuron. SUMOylation plays an integral function in signal transduction. α-amino-3-hydroxy-5-methyl-4-isoxazolepropionic acid receptors (AMPARs) are required for proper synaptic transmission. Although AMPARs are not directly SUMOylated, SUMOylation is required for proper AMPAR trafficking by Arc proteins. N-methyl-D-aspartate receptor (NMDAR) stimulation causes SUMOylation of neuronal nitric oxides synthase (nNOS), which allows signal relay from the nNOS-NO pathway to the extracellular signal-regulated kinase (ERK) pathway. Prevention of nNOS SUMOylation blocks ERK1/2 signaling. Ras SUMOylation is required for signal transduction through the Ras/Raf/MEK/ERK cascade. cAMP response element-binding protein (CREB) SUMOylation increases brain-derived neurotrophic factor (BDNF) levels. BDNF-TrkB signaling activates CREB binding through the ERK pathway. BDNF-TrkB regulates PIAS3 translocation from the nucleus to the dendrites. Downstream of BDNF-TrkB signaling is the P13K-Akt pathway. Akt is SUMOylated at two lysine residues, leading to an increase in Akt activity.

**Figure 3 ijms-23-08023-f003:**
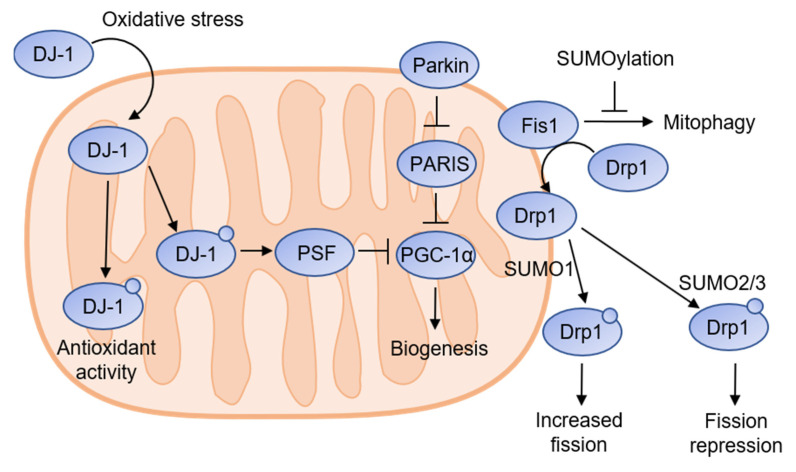
SUMOylation is involved in mitochondrial function. Drp1 is a fission regulatory protein that is found in the cytosol and recruited to the mitochondrial outer membrane by Fis1. Conjugation of SUMO1 to Drp1 increases fission, while SUMO2/3 conjugation represses fission activity. Cellular level of Drp1 is regulated by Parkin E3 ligase, whose activity is regulated by SUMO1 conjugation. Mitophagy is regulated by Fis1, but SUMOylation of Fis1 can repress mitophagy activity. Fis1 levels are regulated by DJ-1, which has antioxidant effects and is recruited into the mitochondria during oxidative stress. SUMOylation is necessary for DJ-1 antioxidant activity. DJ-1 has also been shown to regulate mitochondrial gene expression through indirect control of peroxisome proliferator-activated receptor gamma coactivator 1-alpha (PGC-1α) through polypyrimidine tract-binding protein-associated splicing factor (PSF). DJ-1 inhibits PSF SUMOylation, which increases PSF/PGC-1α binding affinity and suppresses PGC-1α transcriptional activity. PGC-1α levels are regulated by parkin through parkin-interacting substrate (PARIS). PARIS, acting as a suppressor of PGC-1α expression, is regulated by ubiquitin-mediated proteasome via parkin E3 activity. The conjugation of SUMO2/3 to PARIS has been demonstrated to relieve PARIS-mediated transcriptional repression.

## Data Availability

Not applicable.

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
