# Peer review of "SUMOylation and Major Depressive Disorder"

_ijms, 2022, doi:10.3390/ijms23148023_

Round 1
Reviewer 1 Report
Ijms-813017-peer-review-v1
Reviewer Comments
Title: SUMOylation and major depression disorder
Reviewer Comments:
Summary of Manuscript:
This review aimed to provide a summary of recent studies that demonstrate the role of small ubiquitin-like modifiers (SUMO) proteins in the development and maintenance of neuronal dysfunction in individuals with major depressive disorder (MDD). The authors provide several detailed subsections which introduce the topic, elaborate on the SUMOylation process, relate the subunits of α-amino-3-hydroxy-5-methyl-4-isoxazolepropionic acid receptors (AMPARs), and note the parts of the mitochondrial dysfunction. They state that chronic pain has been linked to MDD in humans, and recent studies have shown that SUMOylation is the involved in the regulation of pain in rodent models. The authors conclude that SUMOylation is a promising new field of pain relief and antidepressant medication, and may serve as a potential therapeutic target of MDD in the future.
Overall, the manuscript is worthy of publication, and will make a valuable contribution to the literature in relation to furthering our understanding of the connection between SUMOylation, neuronal dysfunction, MDD, pain, antidepressants, and therapeutic goals. However, there are some minor areas that could be improved (e.g., language, format) before the manuscript is accepted. Please see below for more detailed comments. In many cases, the comments apply throughout the manuscript, so once you make the first correction, you can use it going forwards. I believe these revisions will further strengthen the manuscript.
*Language:
I would like to acknowledge that the authors are from Korea, and have obviously put in a lot of time and effort to translate this manuscript to English. However, there are a few places where the grammar could be improved (see some examples below) for reader clarity:
*Please do a final spell/grammar check for words/phrases in English. For example:
Introduction (line 23) “Changes in synaptic plasticity…effect memory…” (should be “affect”)
Introduction (line 84) “Here we review…proteins associated to…” (should be associated “with”)
Conclusion (line 425) “Mitophagy….insinuating the importance…” (should be “indicating”)
Conclusion (line 441) “Although unconclusive…” (should be “inconclusive”)
*Please revise run-on sentences to improve reader clarity. For example:
“For NMDAR-mediated LTP induction, AMPARs must be concentrated in the postsynaptic membrane, and it is widely believed that this trafficking is possible due to a large number of modifications on AMPAR subunits by PTMs including glycosylation, palmitoylation, phosphorylation and others [41].”
Format:
*Please be clear throughout the Abstract/Text when you are referring to studies with humans, or studies with rodents. There are several paragraphs where you jump back and forth in between different citations (e.g., the first paragraph in the Introduction), and it is unclear to the reader what you are referring to. I suggest separating out the studies with humans in one paragraph, and the studies with rodents in another paragraph, for each subsection as applicable.
*Please consistently use the term of “participants” vs “patients” or “sufferers” throughout the Abstract/Text when you are referring to humans.
*Please consistently use the term “rodent models” vs “rat models” or “mice” throughout the Abstract/Text when you are referring to rodents.
*Please indent each paragraph for ease in readability.
*Please spell out all acronyms at first usage in the Abstract, Text, and Table Footnotes, and then use them consistently throughout (vs. sometimes going back to spelling them out). For example:
SUMO
AMPARS
SENPs
TLD
NMDARs
nNOS
nNOS-NO
MEK
ERK
LTP
RNA
PTMs
ROS
CREB
BDNF
PSF/PGC
PARIS
*Please bold font the Figures citations in the text and the actual Figure titles for ease in readability. Each Figure should be able to stand on its own without the text, so the titles and all variables should be clear (e.g., spelling out all acronyms in the footnotes; see my above note).
*Please provide additional clarifying details for the readers in these areas:
Introduction (line 23): “Causing stress to rats utilizing various methods such as predator and social defeat stress resulted in a sharp increase of H3 phosphorylation and acetylation in the hippocampus.”
*For readers who are not familiar with rodent models, please provide a brief description of “predator and social defeat stress.”
Introduction (line 54): “Major Depressive Disorder (MDD) as defined by the American Psychiatric Association is a common and serious mental disorder that is accompanied by loss of interest and pleasure in enjoyable activities, changes in appetite, loss of energy and other characteristic symptoms.
*Please provide the APA (DSM-5?) citation here, and outline all of the symptoms, vs “other characteristic symptoms.” It is important to give proper credit and clarity of this focal disorder.
Introduction (line 57): “MDD can be potentially fatal with an estimated 1 in 6 people having experienced depression once during their lifetime.”
*Please provide citations for “fatal” and “1 in 6 people.”
Introduction (line 58): “The exact causes of MDD are still elusive, ranging from neural biochemistry to personality. The lack of biomarkers, inconclusive genetics studies and personal effects like stress all contribute to the poor understanding of MDD which also plays a part in antidepressant studies leading to disheartening results.”
* Please provide citations for “neural biochemistry to personality,” “lack of biomarkers, inconclusive genetics studies and personal effects like stress,” and “antidepressant studies leading to disheartening results.”
3.5 SNARE complex (line 227) “…many functions in eukaryotic cells.”
*Please clarify for the reader what “eukaryotic cells” refers to.
3.6 Ion channels (line 254): “By SENP2 knock-out…”
*Please clarify for the reader that “knock-out” refers to rodent models, and briefly describe it.
Conclusion (line 421): “As of late, SUMOylation has been researched in other fields showing promising results. An example would be the control of Fis1 localization in HeLa cells by SENP3-mediated SUMOylation; or localization of mitochondria by mitofusin1/2 mediated by SUMO2 modifications [120,131].”
*Please clarify for the reader which “other fields” you are referring to (e.g., rodent models). If there are further connections to be drawn between these fields, please elaborate on them here.
Conclusion (line 441): “Although unconclusive, recent advances show promising results and suggest that SUMOylation may serve as a potential therapeutic target of MDD in the future.”
*Please relate some examples for the reader in terms of how this therapeutic target could work. The overall research presented here is interesting, however, the reader will want to know more about the clinical implications for humans, and why we should focus our attention on this.
I hope that these comments are helpful, and will assist with further strengthening this manuscript.

Author Response
We thank the referee for the positive overall evaluation and the constructive comments to extend some points in the paper. Please see the attached Word file.

Reviewer 2 Report
Well done for a good job. However, the work need some improvements:
In the abstract, the method through which you searched the literatures should be highlighted. All the credible data bases used such as SCOPUS, Pubmed, Sciencedirect....must be included. The major keywords used for the search should also be included.
In the main body of the review article, the citations need some improvements. As a rule, every first sentence of a paragraph must have a reference.
There are a lot of grammatical errors which you need to check.

Author Response
We thank the reviewer for the positive review and constructive comments. Please see an attached Word file.
